# Employment status and its associated factors for patients 12 months after intensive care: Secondary analysis of the SMAP-HoPe study

Takeshi Unoki[1]*, Mio Kitayama[2], Hideaki Sakuramoto[3], Akira Ouchi[4¤a], Tomoki Kuribara[5¤b], Takako Yamaguchi[6], Sakura Uemura[7], Yuko Fukuda[8], Junpei Haruna[9], Takahiro Tsujimoto[10], Mayumi Hino[11], Yuko Shiba[4], Takumi Nagao[12], Masako Shirasaka[13], Yosuke Satoi[14], Miki Toyoshima[7], Yoshiki Masuda[15], on behalf of the SMAP-HoPe Study Project[¶]

1 Department of Acute and Critical Care Nursing, School of Nursing, Sapporo City University, Sapporo, Japan, 2 Nursing Department Heart Center, Kanazawa Medical University Hospital, Uchinada, Japan, 3 Department of Adult Health Nursing, College of Nursing, Ibaraki Christian University, Hitachi, Japan, 4 Intensive Care Unit, University of Tsukuba Hospital, Tsukuba, Japan, 5 Intensive Care Unit of Advanced Emergency Medical Service Center, Japanese Red Cross Maebashi Hospital, Maebahi, Japan, 6 Intensive Care Unit, Nippon Medical School Musashikosugi Hospital, Kawasaki, Japan, 7 Emergency and Critical Care Medical Center, Osaka City General Hospital, Osaka, Japan, 8 Intensive Care Unit, Jichi Medical University Hospital, Yakushiji Shimotsuke, Japan, 9 Intensive Care Unit, Sapporo Medical University Hospital, Sapporo, Japan, 10 Nursing Practice and Career Support Center, Nara Medical University Hospital, Kashihara, Japan, 11 Intensive Care Unit, Tohoku Medical and Pharmaceutical University Hospital, Sendai, Japan, 12 Intensive Care Unit, Sakakibara Heart Institute, Fuchu, Japan, 13 Intensive Care Unit & Cardiac Care Unit, Japanese Red Cross Fukuoka Hospital, Fukuoka, Japan, 14 Intensive Care Unit, Naha City Hospital, Naha, Japan, 15 Department of Intensive Care Medicine, School of Medicine, Sapporo Medical University, Sapporo, Japan

¤a Current address: Department of Adult Health Nursing, College of Nursing, Ibaraki Christian University, Hitachi, Japan
¤b Current address: Department of Acute and Critical Care Nursing, School of Nursing, Sapporo City University, Chuo-ku, Sapporo, Japan
¶ Membership of the SMAP-HoPe Study Project is provided in the Acknowledgments.
* iwhyh1029@gmail.com

**Data Availability Statement:** All relevant data are within the paper and its Supporting Information files.

## Abstract

### Background

Returning to work is a serious issue that affects patients who are discharged from the intensive care unit (ICU). This study aimed to clarify the employment status and the perceived household financial status of ICU patients 12 months following ICU discharge. Additionally, we evaluated whether there exists an association between depressive symptoms and subsequent unemployment status.

### Methods

This study was a subgroup analysis of the published Survey of Multicenter Assessment with Postal questionnaire for Post-Intensive Care Syndrome for Home Living Patients (the SMAP-HoPe study) in Japan. Eligible patients were those who were employed before ICU admission, stayed in the ICU for at least three nights between October 2019 and July 2020, and lived at home for 12 months after discharge. We assessed the employment status, subjective cognitive functions, household financial status, Hospital Anxiety and Depression

**Funding:** TU received JSPS KAKENHI Grant Number 19K10929. The funders had no role in study design, data collection and analysis, decision to publish, or preparation of the manuscript.

**Competing interests:** The authors have declared that no competing interests exist.

Scale, and EuroQOL-5 dimensions of physical function at 12 months following intensive care.

## Results

This study included 328 patients, with a median age of 64 (interquartile range [IQR], 52–72) years. Of these, 79 (24%) were unemployed 12 months after ICU discharge. The number of patients who reported worsened financial status was significantly higher in the unemployed group (p<0.01) than in the employed group. Multivariable analysis showed that higher age (odds ratio [OR], 1.06; 95% confidence interval [CI], 1.03–1.08) and greater severity of depressive symptoms (OR, 1.13 [95% CI, 1.05–1.23]) were independent factors for unemployment status at 12 months after ICU discharge.

## Conclusions

We found that 24.1% of our patients who had been employed prior to ICU admission were subsequently unemployed following ICU discharge and that depressive symptoms were associated with unemployment status. The government and the local municipalities should provide medical and financial support to such patients. Additionally, community and workplace support for such patients are warranted.

## Introduction

Returning to work following discharge from the intensive care unit (ICU) is a serious issue. A systematic review and meta-analysis of 52 studies on returning to work among previously critically ill patients indicated that delayed return to work and unemployment were common and persistent problems [1]. These studies showed that 36% of patients were subsequently unemployed at 12 months following ICU admission. This change in employment status has a corresponding effect on household income. A study conducted in the United Kingdom suggested that 30% of ICU patients had a decline in household income even after 6 months to 1 year following ICU admission [2]. Additionally, a recent scoping review indicated that 34% of patients that underwent coronary artery bypass grafting or aortic valve replacement surgery never return to work [3].

Return to work is affected by not only the health of the patient but also the environment surrounding the patient. A systematic review of studies on patients with musculoskeletal and pain-related conditions and mental health conditions reported that workplace management was associated with duration of return to work [4]. National employment and disability policies could also contribute to the resumption of work productivity among those discharged from the ICU. Su et al. [5] have revealed that the disability policies of each country were related to the resumption of work productivity. Thus, it is worth evaluating how those discharged from the ICU fare regarding employment in each country. Notably, a prior study in Japan examined the employment status of those discharged from the ICU [6] and found that among 33 patients discharged from the ICU, 26 (83.9%) were able to return to work 6 months following ICU admission. However, the accuracy was insufficient owing to a small sample size. To increase the understanding of employment status among those discharged from the ICU, validation in a larger cohort is therefore warranted.

The factors associated with the unemployment of those discharged from the ICU had been examined in previous studies. Higher age, female sex [7], cognitive function [8], depressive symptoms [9], and physical disability and educational level [10] have been suggested to influence the unemployment status of those discharged from the ICU. However, these factors are inconsistent between studies [1]. Among the general population, absence due to sickness and depression are key factors influencing a longer duration of return to work [11].

Our study primarily aimed to elucidate the employment status and the perceived household financial status among those discharged from the ICU after 12 months. The secondary objective was to determine whether depressive symptoms were associated with subsequent employment status following discharge from intensive care.

## Materials and methods

### Study design

This study was a sub-analysis of the Survey of Multicenter Assessment with Postal questionnaire for Post-Intensive Care Syndrome (PICS) for Home Living Patients (SMAP-HoPe) study [12]. Nested within the SMAP-HoPe study, we conducted an ambidirectional study for patients living at home 12 months following ICU discharge. We sent postal surveys on PICS and employment status. Data from when the patients were admitted were obtained for retrospective analysis. Detailed methods were provided elsewhere in a distinct publication [12].

### Setting

Twelve ICUs in Japan were included in this study. The detailed characteristics of each institution was shown in S1 Table.

### Participants

The study population included patients who stayed in the ICU for at least 3 nights between October 2019 and July 2020 and were living at home 1 year after ICU discharge. Consecutive patients who had been discharged from the ICU 12 months prior were retrospectively enrolled using their medical records at the time of admission, and data on their current employment, economic, and health status were prospectively collected by mail survey. The detailed procedure of the recruitment process was previously published [12]. The number of patients in the previously published study was 754, with a response rate of 91.1%. In this secondary analysis of employment status, only patients aged ≥18 years and those who had been working prior to admission were included.

### Variables/Instruments/Data source

Patients' characteristics, including age, sex, diagnosis, APACHE II score, pre-existing disease, and length of ICU stay, were retrospectively collected by medical chart review. Status after 12 months following discharge from intensive care was collected through postal questionnaires, which explored work status, cognitive function, and Euro-QOL-5D-5L (EQ-5D-5L) [13]. EQ-5D-5L is a validated questionnaire [13], and the Japanese version is available upon request to the EuroQOL group (https://euroqol.org).

The questionnaire on employment status, household financial status, and subjective cognitive function is shown as S1 Text. Work status prior to ICU admission and present working status were classified according to the following scheme: unemployed, self-employed, employed part-time, and employed full-time. Additionally, we inquired about whether household finances changed compared with the period prior to ICU admission. The respondents

selected between a three-point scale of "worse," "no change," or "better." Cognitive function was measured using the following two simple questions that we developed owing to the lack of valid instruments for self-administration: "Do you think your memory function was impaired compared with before hospital admission?" and "Do you think your concentration function was impaired compared with before hospital admission?" Four-point Likert scales, from not at all (0), sometimes (1), frequently (2), and very frequently (3) were the acceptable responses. If the patient responded "very frequently" or "frequently" for either of the two questions, we defined them to have cognitive dysfunction. Usual activities were measured by some questions from the EQ-5D-5L. The response consisted of five levels, from having "no problems" performing usual activities to being "unable to do" their usual activities. We defined responses other than "no problems" or "slight problems" as evidence of physical dysfunction.

Severity of depression was measured by the Hospital Anxiety and Depression Scale (HADS) [14]. HADS comprises depression and anxiety subscales, and each subscale has seven components rated on a scale from 0 to 3. Subscales of depression are graded from 0 to 21. We used the Japanese version of HADS [15], which has been demonstrated to have good reliability and validity.

## Bias

This study was a postal survey, and selection bias may exist; however, selection bias was minimized, given the high response rate in the original study from which this one was nested. Additionally, a state of emergency was declared by the government due to the coronavirus disease 2019 (COVID-19) pandemic in April 2020, which overlapped with the study period. This declaration may have had an impact on employment. This bias was assessed through sensitivity analysis. Furthermore, in several Japanese companies, the compulsory age of retirement was generally between 60 and 65 years. Compulsory retirement, other than the episodes of intensive care admission, may influence our results. Thus, we accordingly conducted sensitivity analysis to evaluate its effect on our findings.

## Sensitivity analysis

We conducted sensitivity analyses to confirm whether our findings were robust. First, we calculated the percentage of unemployment after excluding patients who responded during the COVID-19 pandemic, as employment status after 12 months following ICU discharge may be influenced by the pandemic. It is worth noting that the employment status prior to ICU admission was not considered to be affected by COVID-19 since the respondents were asked to report their status from September 2018 to July 2019. We defined the period after April 2020 as the start of the COVID-19 pandemic because a state of emergency had been declared across the country on April 16, 2020. We calculated the proportion of unemployed patients after excluding those who provided their responses during this period. Second, 60 years old was the beginning of eligibility for compulsory retirement from employment [16]. Thus, we performed multivariable analysis after excluding patients aged ≤60 years and reviewed the impact of retirement age on our findings.

## Statistical analysis

First, descriptive statistical analyses were performed. Continuous variables are expressed as the median (interquartile range [IQR]). The Wilcoxon rank-sum test was used for the comparison of continuous variables. The Chi-square test was used for categorical variables. We calculated the proportion of unemployed subjects stratified by age. For multivariable analysis, a multilevel generalized linear model (GLM) with a binomial distribution and log link was performed to clarify factors for unemployment. We predefined covariates including work status before ICU admission, age [7], sex [7], cognitive dysfunction [8], physical dysfunction [9], and severity of

depression [10] based on previous studies. Additionally, we clarified the relationships between the severity of depression and physical dysfunction. The missing items of the HADS scale were imputed using the "half rule": if half of the items in a subscale were responded to, the mean of the responded scores was imputed [17]. Statistical analysis was performed using STATA IC ver. 16 (Statacorp, TX) and R 4.0.2 (The R foundation for Statistical Computing), and statistical significance was set at $P < 0.05$.

## Ethical considerations

This study was approved by the Human Research Ethics Committee of the Sapporo City University (approval number 1927–1). Additionally, ethical approval was obtained from all centers for the original studies. An explanatory document and consent form were sent to the study participants along with the study set. Participants were instructed to check a box on the consent form to confirm that they understood the study description and agreed to participate; at the three sites, patients confirmed their consent by writing their names on the form, as recommended by the respective institutional review boards.

## Results

### Participants

Fig 1 displays the patient inclusion process. The SMAP-HoPe study had 754 patients. We excluded 425 patients who were unemployed before ICU admission and one patient who did

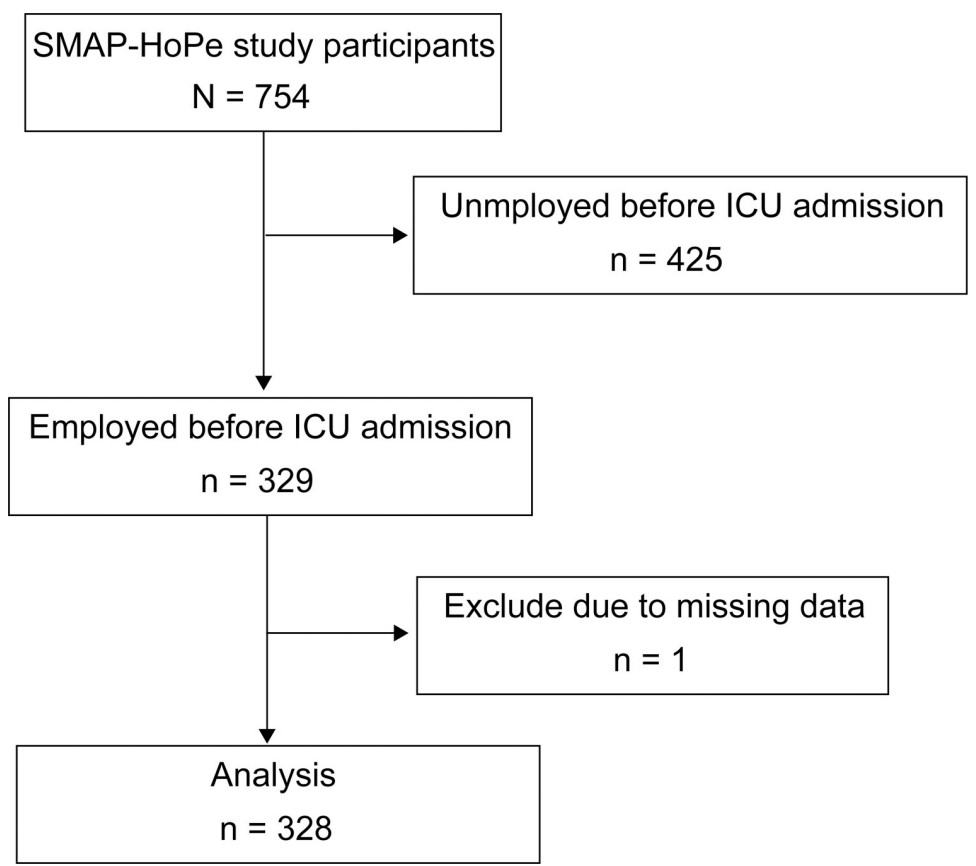

**Fig 1. Patient recruitment scheme.**

not provide an answer to the question of cognitive function. Thus, 328 patients were analyzed in this study. Of the 328 patients, 17 had partial deficits in HADS scores, all of which were imputed by the above method and therefore were not excluded.

## Characteristics of the study population

The characteristics of the study population are shown in Table 1. The median age was 64 (IQR, 52–72) years, with a male predominance (n = 282, 86%). Approximately two fifths of the patients were admitted to the ICU after undergoing elective surgery. The number of patients receiving mechanical ventilation was 219 (66.8%), and the median duration of mechanical ventilation was 2 days. Of the 328 employed patients before ICU admission, the most frequent employment status was full-time employment (n = 156, 47.6%), followed by self-employed (n = 101, 30.8%). The characteristics of each institution, including number of participants and location, are shown in S1 Table.

**Table 1. Demographic characteristics of employed and unemployed subjects 12 months after intensive care.**

| Variables | Participants (n = 328) | Employed (n = 249) | Unemployed (n = 79) | p-value |
|---|---|---|---|---|
| **Age (years), median [IQR]** | 64 [52–72] | 62 [50–71] | 69 [62–75] | <0.001 |
| **Female, n (%)** | 46 (14.0) | 32 (12.9) | 14 (17.7) | 0.271 |
| **Types of admission, n (%)** | | | | |
| **Elective surgery** | 149 (41.2) | 109 (43.8) | 37 (46.2) | 0.697 |
| **Unscheduled admission** | 182 (55.5) | 140 (56.2) | 42 (53.2) | 0.697 |
| **Unscheduled surgery** | 49 (14.9) | 35 (14.1) | 14 (17.7) | 0.469 |
| **Reason for ICU admission, n (%)** | | | | |
| **CV surgery** | 122 (37.2) | 87 (34.9) | 35 (44.3) | 0.183 |
| **CHF/AMI/Arrhy** | 57 (17.4) | 46 (18.5) | 11 (13.9) | |
| **Sepsis** | 34 (10.4) | 21 (8.4) | 13 (16.5) | |
| **Abdominal surgery** | 30 (9.1) | 25 (10.0) | 5 (6.3) | |
| **Other surgery** | 14 (4.3) | 9 (3.6) | 5 (6.3) | |
| **Trauma** | 14 (4.3) | 13 (5.2) | 1 (1.3) | |
| **ENT surgery** | 12 (3.7) | 10 (4.0) | 2 (2.5) | |
| **Aortic dissection (non-operative)** | 12 (3.7) | 11 (4.4) | 1 (1.3) | |
| **Acute renal failure** | 12 (3.7) | 9 (3.6) | 3 (3.8) | |
| **Others** | 21 (6.4) | 18 (7.2) | 3 (3.8) | |
| **Employment status before ICU admission, n (%)** | | | | |
| Self-employed | 101 (30.8) | 86 (34.5) | 15 (19.0) | <0.001 |
| Full time | 156 (47.6) | 125 (50.2) | 31 (39.2) | |
| Part time | 71 (21.6) | 38 (15.3) | 33 (41.8) | |
| **APACHE II, median [IQR]** | 14 [10–19] | 13 [9–19] | 16 [13–19] | 0.001 |
| **MV use, n (%)** | 219 (66.8) | 157 (63.1) | 62 (78.5) | 0.013 |
| **MV (days), median [IQR]** | 2.0 [0.0–3.0] | 2.0 [0.0–3.0] | 2.0 [1.0–3.0] | 0.041 |
| **Psychological history, n (%)** | 4 (1.2) | 4 (1.6) | 0 (0.0) | 0.576 |
| **Delirium (days), median [IQR]** | 0.0 [0.0–1.0] | 0.0 [0.0–1.0] | 0.0 [0.0–2.0] | 0.094 |
| **ICU LOS (days), median [IQR]** | 5.0 [4.0–7.0] | 5.0 [4.0–7.0] | 5.0 [4.0–7.0] | 0.293 |
| **Hospital LOS (days), median [IQR]** | 26.0 [18.0–36.0] | 25.0 [18.0–34.0] | 27.0 [19.0–46.5] | 0.192 |

IQR, interquartile range; CV, cardiovascular; CHF/AMI/arrhy, congestive heart failure/ acute myocardial infarction/arrhythmia; ENT, ear, nose, throat; APACHE II, Acute Physiology and Chronic Health Evaluation II; MV, mechanical ventilation; ICU, intensive care unit; LOS, length of stay; SMAP-HoPe, Survey of Multicenter Assessment with Postal questionnaire for Post-Intensive Care Syndrome (PICS) for Home Living Patients.

## Characteristics of employed and unemployed patients

Among the 328 patients who were previously employed, 79 (24.1%; 95% confidence interval [CI], 19.7–29.1) were unemployed at 12 months following ICU discharge. A comparison of the patient characteristics between those employed and those unemployed after ICU discharge is shown in Table 1. The median age of those employed was significantly lower than that of those unemployed (62 years [50–71] vs. 69 years [62–75], p<0.01, respectively). At older ages, the percentage of unemployed patients was higher (S2 Table). The median APACHE II score of employed patients was lower than that of unemployed patients (13 [9–19] vs. 16 [13–19], p<0.001, respectively). The reasons for ICU admission between the employed and unemployed groups following intensive care was comparable (p = 0.183).

## Employment status

Fig 2 shows the employment status between the period prior to ICU admission and 12 months following ICU discharge. Prior to ICU admission, there were 156 full-time employees. Of those, 31 (19.9%) were unemployed at 12 months following ICU discharge. Additionally, of the 71 patients with part-time employment, 33 (46.5%) were unemployed at 12 months following ICU discharge.

## Household financial status

Fig 3 shows the proportion of respondents stratified by employment status 12 months following ICU discharge who perceived lower household financial status. Approximately half of those unemployed perceived a worse financial status compared with the period prior to ICU admission (p<0.001).

The number of patients who perceived their home finances as worse 12 months after ICU discharge than before ICU admission was significantly higher among patients who were unemployed compared to those who were employed (p<0.001).

## Multivariable analysis

Multilevel GLM showed that higher age, part-time or self-employed status prior to ICU admission, and greater severity of depressive symptoms were independent factors for unemployment status at 12 months following admission. The results of the multivariable analysis are shown in Table 2.

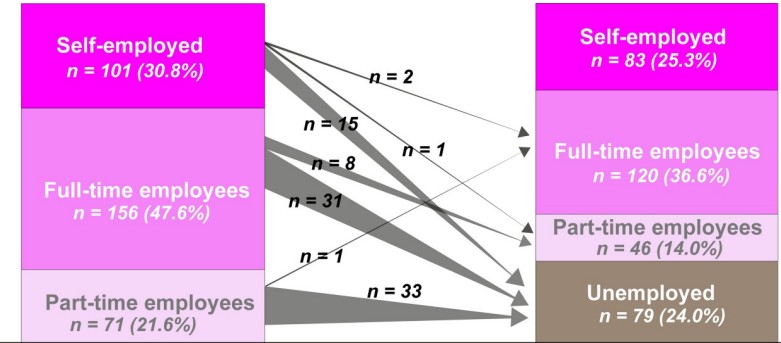

**Fig 2. Change in work status before ICU admission and 12 months after ICU discharge.**

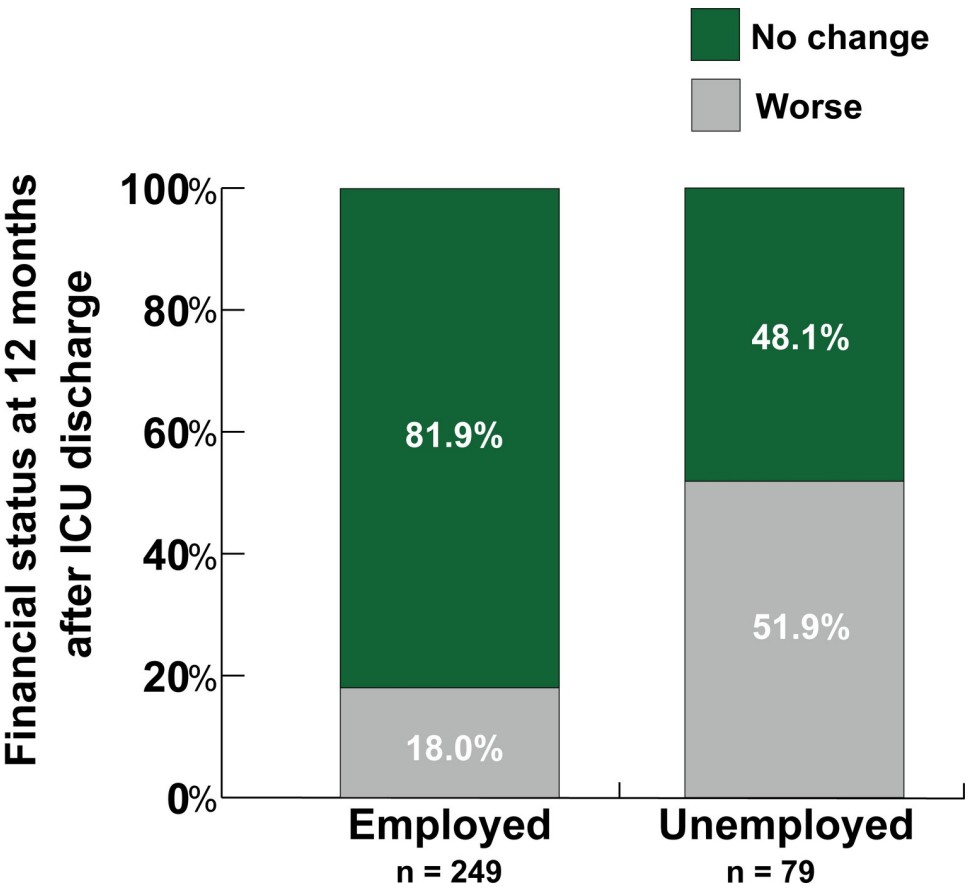

**Fig 3. Perceived change in household financial status before and after ICU admission.**

### Relationships between severity of depression and physical dysfunction

The median severity of depression score was significantly higher in those with physical dysfunction than in those without physical dysfunction (8 [6–10] vs. 4 [2–7], p<0.01, respectively).

### Sensitivity analysis

First, we calculated the proportion of the unemployed after excluding all patients who responded within the duration of the COVID-19 pandemic. There were 178 patients who were

**Table 2. Multivariable analysis of factors associated with unemployed status 12 months after intensive care unit discharge.**

| Variable | Odds ratio | 95% CI | p-value |
|---|---|---|---|
| Age | 1.06 | 1.03–1.08 | <0.001 |
| Male | 0.70 | 0.31–1.58 | 0.393 |
| Previous employment status | | | |
| Part-time employed[a] | 2.28 | 1.16–4.48 | 0.017 |
| Self-employed[a] | 0.27 | 0.12–0.60 | <0.001 |
| Cognitive impairment | 1.08 | 0.39–2.95 | 0.886 |
| Physical dysfunction | 2.43 | 0.92–6.40 | 0.073 |
| Severity of depression | 1.13 | 1.05–1.23 | 0.003 |

[a]Full-time employment as a reference.

employed during the period prior to ICU admission and 40 unemployed patients. The unemployment rate was 22.4% (95% CI, 16.9–29.2). Second, we excluded those aged ≤60 years and conducted multilevel GLM following the same procedure as the primary analysis. Consequently, age was not determined to be an independent factor; however, worse depressive symptoms remained an independent factor among the unemployed (S3 Table).

## Discussion

We determined that one-fourth of the patients who were previously employed prior to ICU admission were subsequently unemployed at 12 months following ICU discharge. Additionally, over half of the unemployed patients perceived worse household financial status compared to that in the period prior to ICU admission. Severity of depressive symptoms, higher age, and part-time or self-employed status prior to ICU admission were identified as independent factors for unemployment.

In this study, one-fourth of the patients who were previously employed prior to ICU admission were subsequently unemployed at 12 months following ICU discharge. A recent meta-analysis reported that 40% of patients did not return to work at 12 months following discharge from intensive care [1]. Compared with the aforementioned study [1], we found a relatively lower rate of unemployment. We attribute this discrepancy to three reasons.

First is the characteristics of patients admitted to the ICU. A previous study reported that 59.2% of patients in Japan entered the ICU after elective surgery compared with 42.3% in the USA and that fewer patient entered the ICU from the emergency department in Japan [18]. In the present study, 41.2% of the patients were also admitted to the ICU for elective surgery. This may be one of the reasons for the difference between our study and previous studies [1].

Second, although it has not always been the case in recent years, there was a "lifetime employment system," which may be specific to Japan. Under this system, employees up to their 50s tend not to be laid off [19]. This may influence our findings because our study population had a median age of 64 years.

Third, disability policy in each country may influence the proportion of employees returning to work among patients discharged from intensive care. A recent systematic review and meta-analysis using meta-regression revealed that disability policies in each country positively affected the ability of ICU-discharged patients to return to work [5]. Based on the Organization for Economic Co-operation and Development (OECD) data, the integration index calculated from ten criteria, such as employment programs and government-provided job training, was higher in Japan than the median index of OECD [20].

Depressive symptoms are possibly associated with unemployment status after ICU discharge. We found that depressive symptoms were an independent factor associated with unemployment despite adjusting for covariates, including physical function and cognitive function. These findings are consistent with our hypothesis. A previous study suggested that depression was associated with not returning to work in Australia; however, the study did not conduct multivariable analysis [21]. The present study did not indicate causal relationships between unemployment and depression. Depression may lead to unemployment and vice versa from our study design.

Patients with physical dysfunction had a greater severity of depression; however, physical dysfunction was not an independent factor for unemployment status as indicated by our multivariable analysis results. A study conducted in PICS clinic indicated that patients with walking disability had a greater severity of depression than patients without walking disability after intensive care [22]. Additionally, the study has shown that higher grip strength was associated

with a higher total HADS score [22]. Thus, physical dysfunction may be associated with a greater severity of depression.

Our study did not identify physical dysfunction as an independent factor for unemployment status. This finding was consistent with that of a previous study reporting that physical dysfunction was independently associated with employment status 3 months after intensive care and not a year later [10]. Therefore, physical function may have a significant impact on return to work in the short term after ICU discharge; however, the impact may become relatively negligible in the long term. As the present study was conducted on patients 1 year after ICU discharge, the influence of physical function may have been relatively small.

## Strengths of the study

We considered that the patients analyzed in this study are representative of ICU patients admitted in Japan. Compared to that in the Japanese Intensive Care Patient Database report [23], a database of ICUs in 21 Japanese institutions, the median age of the patients in this study was lower; however, other variables, including the proportion of those who underwent elective surgery, cardiovascular surgery, and severity of illness were comparable. Additionally, the study was obtained from 12 ICUs in a wide geographic area and from a variety of hospital types, such as university hospitals and public hospitals, in Japan.

## Limitations of the study

First, since we simultaneously collected data on employment status and depressive symptoms, we were not able to clarify the causal relationship between these two variables. Further research is needed to assess depressive symptoms at discharge and subsequently assesses return to work status.

Second, the COVID-19 pandemic may affect our findings. Approximately half of the mail survey period overlapped with the pandemic; however, the level of spread was different in the region. Because of the ambidirectional study design, there was no effect of COVID-19 on patients' employment status and household finances prior to ICU admission; however, employment and household finances at 12 months following ICU discharge were likely influenced by the COVID-19 pandemic. We considered this influence as minimal, since our sensitivity analysis showed similar results despite the exclusion of subjects who responded during the COVID-19 pandemic.

Third, we included patients of all ages; thus, retirement age affected our results. We consider this effect inconsequential to our findings based on results of our sensitivity analysis.

Fourth, to the best of our knowledge, there were few translated and validated self-administered questionnaires in Japanese, and we were not able to use a valid tool to measure cognitive function.

Fifth, we excluded patients with central nervous system disease as this study was based on self-administered questionnaires. Thus, our findings may underestimate the proportion of the unemployed.

## Implications for future research

It is likely that the proportion of those returning to work depends not only on mental health or physical function but also employment culture and governmental policy. This study was a subgroup analysis; thus, studies with a larger cohort focusing on return to work following intensive care admission are warranted for each country.

### Implications for clinical practice

Patients discharged from the ICU should be carefully monitored, even if they are at home. In particular, depression is associated with unemployment status and should be monitored. The government and local municipalities should provide medical and financial support to such patients. Additionally, support in the workplace will be essential to facilitate return to work. Workplace support may require multi-component interventions, including health-related support, service coordination including return-to-work planning and case management, and work modifications including modifications for working hours and duties [4].

## Conclusion

We found that 24.1% of our patients who had been employed prior to ICU admission were subsequently unemployed 12 months after ICU discharge. Additionally, depressive symptoms were associated with unemployment status. Employment status and mental health should be followed up, and adequate support is warranted.

## Supporting information

**S1 Table. Characteristics and number of analyzed participants for each intensive care unit.**
(DOCX)

**S2 Table. Distribution of age group and proportion of unemployed subjects stratified by age group.**
(DOCX)

**S3 Table. Sensitivity analysis: Multivariable analysis of factors associated with unemployed status 12 months after intensive care unit discharge among patients after excluding patients aged 60 years or younger.**
(DOCX)

**S1 Text. Questionnaire regarding employment, household finance, and subjective cognitive function.**
(DOCX)

## Acknowledgments

The following SMAP-HoPe Study Project investigators were involved in the protocol: Ryuta Indo, Hiroomi Tatsumi, Atsuko Handa, Kazuyo Koori, Ayano Kudo, Kayo Kitaura, Etsuko Moro, Shin Nunomiya, Akira Ouchi, Masako Sato, Yoshiaki Inoue, Etsuko Tsukioka, Yasuhiro Kishi, Chiaki Fujii, Kohei Matsuba, Hiroki Isonishi, Ikumi Kobashi, Miki Toyoshima, Masahiro Yamane, Yumi Kajiyama, and Yoshifumi Heshiki.

## Author Contributions

**Conceptualization:** Takeshi Unoki, Hideaki Sakuramoto.

**Data curation:** Takeshi Unoki.

**Formal analysis:** Takeshi Unoki, Mio Kitayama, Akira Ouchi.

**Funding acquisition:** Takeshi Unoki.

**Investigation:** Takeshi Unoki, Mio Kitayama, Akira Ouchi, Tomoki Kuribara, Takako Yamaguchi, Sakura Uemura, Yuko Fukuda, Junpei Haruna, Takahiro Tsujimoto, Mayumi Hino,

Yuko Shiba, Takumi Nagao, Masako Shirasaka, Yosuke Satoi, Miki Toyoshima, Yoshiki Masuda.

**Methodology:** Takeshi Unoki, Hideaki Sakuramoto.

**Project administration:** Takeshi Unoki.

**Visualization:** Takeshi Unoki.

**Writing – original draft:** Takeshi Unoki, Mio Kitayama, Hideaki Sakuramoto, Tomoki Kuribara.

**Writing – review & editing:** Takeshi Unoki, Mio Kitayama, Hideaki Sakuramoto, Akira Ouchi, Tomoki Kuribara, Takako Yamaguchi, Sakura Uemura, Yuko Fukuda, Junpei Haruna, Takahiro Tsujimoto, Mayumi Hino, Yuko Shiba, Takumi Nagao, Masako Shirasaka, Yosuke Satoi, Miki Toyoshima, Yoshiki Masuda.

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
