## [Decision Letter · Decision Letter 0]

22 Dec 2021

PONE-D-21-23032Employment status and its associated factor for patients 12 months after intensive care: Secondary analysis of the SMAP-HoPe-studyPLOS ONE

Dear Author,

Thank you for submitting your manuscript to PLOS ONE. After careful consideration, we feel that it has merit but does not fully meet PLOS ONE’s publication criteria as it currently stands. Therefore, we invite you to submit a revised version of the manuscript that addresses the points raised during the review process.

ACADEMIC EDITORJournal Requirements:

2. It is imperative to amend the abstract section to structured (i.e Introduction, Methods, result, conclusion) as per PLOS ONE's style requirements.

We look forward to receiving your revised manuscript.

Kind regards,

Gebisa Guyasa Kabito, MPH

Academic Editor

PLOS ONE

Journal Requirements:

“TU received JSPS KAKENHI Grant Number 19K10929. The funders had no role in study design, data collection and analysis, decision to publish, or preparation of the manuscript”

Additional Editor Comments (if provided):

In your Methods section, please provide additional information about the participant recruitment method and the demographic details of your participants.

Please ensure you have provided sufficient details to replicate the analyses such as:

a) a statement as to whether your sample can be considered representative of a larger population, and

b) a description of how participants were recruited.

Reviewers' comments:

Reviewer's Responses to Questions

**Comments to the Author**

1. Is the manuscript technically sound, and do the data support the conclusions?

Reviewer #1: Yes

2. Has the statistical analysis been performed appropriately and rigorously? 

Reviewer #1: Yes

3. Have the authors made all data underlying the findings in their manuscript fully available?

Reviewer #1: Yes

4. Is the manuscript presented in an intelligible fashion and written in standard English?

Reviewer #1: Yes

5. Review Comments to the Author

Reviewer #1: The paper presented concerns some relevant and interesting aspects of the returning to work of patients admitted to the ICU. The paper is well written and underlines the importance of the mental wellbeing to assure the patients a successfull return to work. The manuscript is well organized and the findings are clearly exposed.

Here are some questions that I would like the authors to address.

-is there any other illness apart from depression identified in the ICU patients that can determine a delay in the return to work or prevent the patients to the return to work?

-Does the physical function influenced the return to work and was it associated to the depression status?

Moreover, I think that in the Introduction paragraph, some insights are needed about the return to work of workers affected by serious illnesses and about the mangement of their special needs in the workplace. Please check the following references that can be useful for the insight:

-Cullen KL, Irvin E, Collie A, Clay F, Gensby U, Jennings PA, Hogg-Johnson S, Kristman V, Laberge M, McKenzie D, Newnam S, Palagyi A, Ruseckaite R, Sheppard DM, Shourie S, Steenstra I, Van Eerd D, Amick BC 3rd. Effectiveness of Workplace Interventions in Return-to-Work for Musculoskeletal, Pain-Related and Mental Health Conditions: An Update of the Evidence and Messages for Practitioners. J Occup Rehabil. 2018 Mar;28(1):1-15. doi: 10.1007/s10926-016-9690-x. PMID: 28224415; PMCID: PMC5820404.

-Tan FSI, Shorey S. Experiences of women with breast cancer while working or returning to work: a qualitative systematic review and meta-synthesis. Support Care Cancer. 2021 Oct 13. doi: 10.1007/s00520-021-06615-w. Epub ahead of print. PMID: 34647131.

-Mortensen M, Sandvik RKNM, Svendsen ØS, Haaverstad R, Moi AL. Return to work after coronary artery bypass grafting and aortic valve replacement surgery: A scoping review. Scand J Caring Sci. 2021 May 31. doi: 10.1111/scs.13006. Epub ahead of print. PMID: 34057755.

-Mecheri, V., Fioriti, M., Lulli, L. G., Taddei, G., Fiz Perez, J., & Cupelli, V. (2019). Management strategies for the occupational reintegration of the worker with cardiovascular disease. Quality - Access to Success, 20(171), 157-162. Retrieved from www.scopus.com

-Hodgson CL, Higgins AM, Bailey MJ, Mather AM, Beach L, Bellomo R, Bissett B, Boden IJ, Bradley S, Burrell A, Cooper DJ, Fulcher BJ, Haines KJ, Hopkins J, Jones AYM, Lane S, Lawrence D, van der Lee L, Liacos J, Linke NJ, Gomes LM, Nickels M, Ntoumenopoulos G, Myles PS, Patman S, Paton M, Pound G, Rai S, Rix A, Rollinson TC, Sivasuthan J, Tipping CJ, Thomas P, Trapani T, Udy AA, Whitehead C, Hodgson IT, Anderson S, Neto AS; COVID-Recovery Study Investigators and the ANZICS Clinical Trials Group. The impact of COVID-19 critical illness on new disability, functional outcomes and return to work at 6 months: a prospective cohort study. Crit Care. 2021 Nov 8;25(1):382. doi: 10.1186/s13054-021-03794-0. PMID: 34749756; PMCID: PMC8575157.

6. PLOS authors have the option to publish the peer review history of their article (what does this mean?). If published, this will include your full peer review and any attached files.

Reviewer #1: No

---

## [Author Response · Author response to Decision Letter 0]

11 Jan 2022

January 11th, 2022

Dr. Emily Chenette

Editor-in-Chief

PLoS One

Dear Dr. Emily Chenette:

I, along with my coauthors, would like to re-submit the attached manuscript entitled “Employment status and its associated factor for patients 12 months after intensive care: Secondary analysis of the SMAP-HoPe-study” for publication in PLoS One as an original research article. The manuscript ID is PONE-D-21-23032.

We thank academic editor and the reviewer for the consideration that you have accorded our manuscript. The academic editor and reviewer’s comments have helped us improve our manuscript, and we are grateful for all the suggestions. We have addressed all the academic editor and reviewer’s comments separately below.

Additionally, we have added a description of the handling of missing values. We have also added the number of participants at each site to the Supporting Information. In addition, we used an English editing service to improve the overall English. The page and line numbers followed the without track-change version.

We hope that the revised manuscript is now suitable for publication in your journal.

Sincerely,

Takeshi Unoki

Department of Acute and Critical Care Nursing, School of Nursing, Sapporo City University

060-0011

Kita 11 Nishi 13, Chuo-ku, Sapporo, Hokkaido, Japan

Phone & Fax: +81-11-726-2557

E-mail: iwhyh1029@gmail.com

 

Response to Editor and Reviewer

Comment

Journal Requirements:

Response

Thank you for pointing this out. We have made the corrections as per PLOS ONE’s style requirements.

Comment

Response

We have checked the statement of research grant and will respond to it along with point 3.

Comment

“TU received JSPS KAKENHI Grant Number 19K10929. The funders had no role in study design, data collection and analysis, decision to publish, or preparation of the manuscript”

Response

We have removed "Financial Disclosure" from the manuscript and revised the funding information. The revised text is included in the cover letter.

Comment

4. Please review your reference list to ensure that it is complete and correct. If you have cited papers that have been retracted, please include the rationale for doing so in the manuscript text or remove these references and replace them with relevant current references. Any changes to the reference list should be mentioned in the rebuttal letter that accompanies your revised manuscript. If you need to cite a retracted article, indicate the article’s retracted status in the References list and also include a citation and full reference for the retraction notice.

Response

We have confirmed that there were no retracted articles. We have indicated the revised reference list at the end of this letter.

Comment

Response

Our developed questionnaire regarding employment, household finance, and subjective cognitive functions was additionally shown in S1 text.

Additional Editor Comments (if provided):

Comment

In your Methods section, please provide additional information about the participant recruitment method and the demographic details of your participants.

Please ensure you have provided sufficient details to replicate the analyses such as:

Response

Thank you for your comment. We have added descriptions regarding the recruitment method and characteristics of the participants. We will respond with the following comments regarding the changes.

Comment

a) a statement as to whether your sample can be considered representative of a larger population

Response

We have added the description if we considered the samples; the analysis was representive of a larger population in the Discussion section. The following sentences in red font were added. 

 ------Revised Manuscript------

P19, Line 367

Strengths of the study

We considered that the patients analyzed in this study are representative of ICU patients admitted in Japan. Compared to that in the Japanese Intensive Care Patient Database report [23], a database of ICUs in 21 Japanese institutions, the median age of the patients in this study was lower; however, other variables, including the proportion of those who underwent elective surgery, cardiovascular surgery, and severity of illness were comparable. Additionally, the study was obtained from 12 ICUs in a wide geographic area and from a variety of hospital types, such as university hospitals and public hospitals, in Japan. 

Additionally, we have added supporting information including the type of institution, number of ICU beds, and location of each facility where the recruitment took place. These were added to S1 table.　We hope this information helps the reader understand that samples are taken from populations with a wide variety in terms of geography and installation entities. The following sentence in red font was added to the manuscript in the result section.

------Revised Manuscript------

P12, Line 240

The number of patients receiving mechanical ventilation was 219 (66.8%), and the median duration of mechanical ventilation was 2 days. Of the 328 employed patients before ICU admission, the most frequent employment status was part-time employment (n=125, 50.2%), followed by full-time employment (n=86, 34.5%). The characteristics of each institution, including number of participants and location, are shown in S2 Table. 

Comment

b) a description of how participants were recruited.

Response

We have added the description on how we recruited the participants. The following red sentences were added.

 ------Revised Manuscript------

P7, Line 128

Participants 

The study population included patients who stayed in the ICU for at least 3 nights between October 2019 and July 2020 and were living at home 1 year after ICU discharge. Consecutive patients who had been discharged from the ICU 12 months prior were retrospectively enrolled using their medical records at the time of admission, and data on their current employment, economic, and health status were prospectively collected by mail survey. The detailed procedure of the recruitment process was previously published [12]. The number of patients in the previously published study was 754, with a response rate of 91.1%. In this secondary analysis of employment status, only patients aged ≥18 years and those who had been working prior to admission were included.

 

Reviewers' comments:

Reviewer's Responses to Questions

Comments to the Author

Comment

Reviewer #1: The paper presented concerns some relevant and interesting aspects of the returning to work of patients admitted to the ICU. The paper is well written and underlines the importance of the mental wellbeing to assure the patients a successfull return to work. The manuscript is well organized and the findings are clearly exposed.

Here are some questions that I would like the authors to address.

-is there any other illness apart from depression identified in the ICU patients that can determine a delay in the return to work or prevent the patients to the return to work?

Response

Thank you for the comment. In this analysis, other illness aside from depression were not associated with employment status. Because we believe that covariates should be predefined, we are unable to add illness such as sepsis as covariates. However, in univariable analysis, diagnosis at ICU of patients between the employed group and unemployed group following intensive care was comparable. We speculated that illness at the ICU would not significantly influence employment status 12 months after intensive care.

We have added the statement that illness during ICU did not statistically differ between the two groups in univariable analysis in the result section.

The following red sentences were added.

------Revised Manuscript------

P12, Line 243

Among the 328 patients who were previously employed, 79 (24.1%; 95% confidence interval [CI], 19.7–29.1) were unemployed at 12 months following ICU discharge. A comparison of the patient characteristics between those employed and those unemployed after ICU discharge is shown in Table 1. The median age of those employed was significantly lower than that of those unemployed (62 years [50–71] vs. 69 years [62–75], p<0.01, respectively). At older ages, the percentage of unemployed patients was higher (S2 Table). The median APACHE II score of employed patients was lower than that of unemployed patients (13 [9–19] vs. 16 [13–19], p<0.001, respectively). The reasons for ICU admission between the employed and unemployed groups following intensive care was comparable (p=0.183).

Comment

-Does the physical function influenced the return to work and was it associated to the depression status?

Response

We additionally analyzed the relationship between severity depressive symptom and physical dysfunction. Impaired physical function was associated with higher severity of depressive symptom. Physical dysfunction was not an independent factor for employment status after adjusted covariates including severity of depressive symptom. 

We have added the statement that we planned to analyze the relationship between severity of depressive symptom and physical function in the method section.

------Revised Manuscript------

P10, Line 202

For multivariable analysis, a multilevel generalized linear model (GLM) with a binomial distribution and log link was performed to clarify factors for unemployment. We predefined covariates including work status before ICU admission, age [7], sex [7], cognitive dysfunction [8], physical dysfunction [9], and severity of depression [10] based on previous studies. Additionally, we clarified the relationships between the severity of depression and physical dysfunction.

Additionally, we have added a description of the results in the results section.

------Revised Manuscript------

P16, Line 295

Relationships between severity of depression and physical dysfunction

The median severity of depression score was significantly higher in those with physical dysfunction than in those without physical dysfunction (8 [6–10] vs 4 [2–7], p<0.01, respectively).

Moreover, we have added the description interpretation of the results in the discussion section.

------Revised Manuscript------

P18, Line 342

Depressive symptoms are possibly associated with unemployment status after ICU discharge. We found that depressive symptoms were an independent factor associated with unemployment despite adjusting for covariates, including physical function and cognitive function. These findings are consistent with our hypothesis. A previous study suggested that depression was associated with not returning to work in Australia; however, the study did not conduct multivariable analysis [21]. The present study did not indicate causal relationships between unemployment and depression. Depression may lead to unemployment and vice versa from our study design.

Patients with physical dysfunction had a greater severity of depression; however, physical dysfunction was not an independent factor for unemployment status as indicated by our multivariable analysis results. A study conducted in PICS clinic indicated that patients with walking disability had a greater severity of depression than patients without walking disability after intensive care [22]. Additionally, the study has shown that higher grip strength was associated with a higher total HADS score [22]. Thus, physical dysfunction may be associated with a greater severity of depression.

Our study did not identify physical dysfunction as an independent factor for unemployment status. This finding was consistent with that of a previous study reporting that physical dysfunction was independently associated with employment status 3 months after intensive care and not a year later [10]. Therefore, physical function may have a significant impact on return to work in the short term after ICU discharge; however, the impact may become relatively negligible in the long term. As the present study was conducted on patients 1 year after ICU discharge, the influence of physical function may have been relatively small.

Comment

I think that in the Introduction paragraph, some insights are needed about the return to work of workers affected by serious illnesses and about the mangement of their special needs in the workplace. Please check the following references that can be useful for the insight:

Response

Thank you for the new insight. We now understand that workplace management has a great impact on return to work, and we have added references to this perspective in the Introduction.

------Revised Manuscript------

P5, Line 79

Introduction

Returning to work following discharge from the intensive care unit (ICU) is a serious issue. A systematic review and meta-analysis of 52 studies on returning to work among previously critically ill patients indicated that delayed return to work and unemployment were common and persistent problems [1]. These studies showed that 36% of patients were subsequently unemployed at 12 months following ICU admission. This change in employment status has a corresponding effect on household income. A study conducted in the United Kingdom suggested that 30% of ICU patients had a decline in household income even after 6 months to 1 year following ICU admission [2]. Additionally, a recent scoping review indicated that 34% of patients that underwent coronary artery bypass grafting or aortic valve replacement surgery never return to work [3].

Return to work is affected by not only the health of the patient but also the environment surrounding the patient. A systematic review of studies on patients with musculoskeletal and pain-related conditions and mental health conditions reported that workplace management was associated with duration of return to work [4]. National employment and disability policies could also contribute to the resumption of work productivity among those discharged from the ICU.

We believe that this perspective is significant and have decided to add it to the implications for clinical practice.

------Revised Manuscript------

P21, Line 406

Implications for clinical practice

Patients discharged from the ICU should be carefully monitored, even if they are at home. In particular, depression is associated with unemployment status and should be monitored. The government and local municipalities should provide medical and financial support to such patients. Additionally, support in the workplace will be essential to facilitate return to work. Workplace support may require multi-component interventions, including health-related support, service coordination including return-to-work planning and case management, and work modifications including modifications for working hours and duties [4].

References

Articles indicated in red font were added after the first submission.

1. Kamdar BB, Suri R, Suchyta MR, Digrande KF, Sherwood KD, Colantuoni E, et al. Return to work after critical illness: a systematic review and meta-analysis. Thorax. 2020;75: 17-27. doi:10.1136/thoraxjnl-2019-213803

2. Griffiths J, Hatch RA, Bishop J, Morgan K, Jenkinson C, Cuthbertson BH, et al. An exploration of social and economic outcome and associated health-related quality of life after critical illness in general intensive care unit survivors: a 12-month follow-up study. Crit Care. 2013;17: R100. doi:10.1186/cc12745

3. Mortensen M, Sandvik RKNM, Svendsen ØS, Haaverstad R, Moi AL. Return to work after coronary artery bypass grafting and aortic valve replacement surgery: A scoping review. Scand J Caring Sci. 2021. doi:10.1111/scs.13006

4. Cullen KL, Irvin E, Collie A, Clay F, Gensby U, Jennings PA, et al. Effectiveness of workplace interventions in return-to-work for musculoskeletal, pain-related and mental health conditions: An update of the evidence and messages for practitioners. J Occup Rehabil. 2018;28: 1-15. doi:10.1007/s10926-016-9690-x

5. Su H, Dreesmann NJ, Hough CL, Bridges E, Thompson HJ. Factors associated with employment outcome after critical illness: Systematic review, meta-analysis, and meta-regression. J Adv Nurs. 2021;77: 653-663. doi:10.1111/jan.14631

6. Kawakami D, Fujitani S, Morimoto T, Dote H, Takita M, Takaba A, et al. Prevalence of post-intensive care syndrome among Japanese intensive care unit patients: a prospective, multicenter, observational J-PICS study. Crit Care. 2021;25: 69. doi:10.1186/s13054-021-03501-z

7. Myhren H, Ekeberg Ø, Stokland O. Health-related quality of life and return to work after critical illness in general intensive care unit patients: a 1-year follow-up study. Crit Care Med. 2010;38: 1554–1561. doi:10.1097/CCM.0b013e3181e2c8b1

8. Rothenhäusler HB, Ehrentraut S, Stoll C, Schelling G, Kapfhammer HP. The relationship between cognitive performance and employment and health status in long-term survivors of the acute respiratory distress syndrome: results of an exploratory study. Gen Hosp Psychiatry. 2001;23: 90–96. doi:10.1016/s0163-8343(01)00123-2

9. Zisopoulos G, Roussi P, Mouloudi E. Psychological morbidity a year after treatment in intensive care unit. Health Psychol Res. 2020;8: 8852. doi:10.4081/hpr.2020.8852

10. Norman BC, Jackson JC, Graves JA, Girard TD, Pandharipande PP, Brummel NE, et al. Employment outcomes after critical illness: an analysis of the bringing to light the risk factors and incidence of neuropsychological dysfunction in ICU survivors cohort. Crit Care Med. 2016;44: 2003–2009. doi:10.1097/CCM.0000000000001849

11. Vlasveld MC, van der Feltz-Cornelis CM, Bültmann U, Beekman ATF, van Mechelen W, Hoedeman R, et al. Predicting return to work in workers with all-cause sickness absence greater than 4 weeks: a prospective cohort study. J Occup Rehabil. 2012;22: 118-126. doi:10.1007/s10926-011-9326-0

12. Unoki T, Sakuramoto H, Uemura S, Tsujimoto T, Yamaguchi T, Shiba Y, et al. Prevalence of and risk factors for post-intensive care syndrome: Multicenter study of patients living at home after treatment in 12 Japanese intensive care units, SMAP-HoPe study. PLoS One. 2021;16: e0252167. doi:10.1371/journal.pone.0252167

13. Herdman M, Gudex C, Lloyd A, Janssen M, Kind P, Parkin D, et al. Development and preliminary testing of the new five-level version of EQ-5D (EQ-5D-5L). Qual Life Res. 2011;20: 1727-1736. doi:10.1007/s11136-011-9903-x

14. Zigmond AS, Snaith RP. The hospital anxiety and depression scale. Acta Psychiatr Scand. 1983;67: 361–370. doi:10.1111/j.1600-0447.1983.tb09716.x

15. Hatta H, Higashi A, Yashiro H, Kotaro O, Hayashi K, Kiyota K, et al. A validation of the hospital anxiety and depression scale. Jpn J Psychosom Med. 1998;38: 309-315. doi:10.15064/jjpm.38.5_309

16. Ministry of Health, Labour and Welfare. Overview of the results of the Comprehensive Survey on Working Conditions in 2017. 27 Dec 2017 [cited 2020 Dec 30]. Available from: https://www.mhlw.go.jp/toukei/itiran/roudou/jikan/syurou/17/

17. Bell ML, Fairclough DL, Fiero MH, Butow PN. Handling missing items in the Hospital Anxiety and Depression Scale (HADS): a simulation study. BMC Res Notes. 2016;9: 479. doi:10.1186/s13104-016-2284-z

18. Sirio CA, Tajimi K, Taenaka N, Ujike Y, Okamoto K, Katsuya H. A cross-cultural comparison of critical care delivery: Japan and the United States. Chest. 2002;121: 539-548. doi:10.1378/chest.121.2.539

19. Ono H. Lifetime employment in Japan: Concepts and measurements. J Jpn Int Econ. 2010;24: 1–27. doi:10.1016/j.jjie.2009.11.003

20. Organisation for Economic Co-operation and Development. Sickness, disability and work breaking the barriers : a synthesis of findings across OECD countries. Paris: OECD; 2010. Available from: https://www.worldcat.org/title/sickness-disability-and-work-breaking-the-barriers-a-synthesis-of-findings-across-oecd-countries/oclc/1136220722?referer=di&ht=edition

21. Hodgson CL, Haines KJ, Bailey M, Barrett J, Bellomo R, Bucknall T, et al. Predictors of return to work in survivors of critical illness. J Crit Care. 2018;48: 21–25. doi:10.1016/j.jcrc.2018.08.005

22. Nakamura K, Kawasaki A, Suzuki N, Hosoi S, Fujita T, Hachisu S, et al. Grip strength correlates with mental health and quality of life after critical care: a retrospective study in a post-intensive care syndrome clinic. J Clin Med Res. 2021;10. doi:10.3390/jcm10143044

23. Irie H, Okamoto H, Uchino S, Endo H, Uchida M, Kawasaki T, et al. The Japanese Intensive care PAtient Database (JIPAD): A national intensive care unit registry in Japan. J Crit Care. 2020;55: 86-94. doi:10.1016/j.jcrc.2019.09.004

---

## [Editor Report · Decision Letter 1]

20 Jan 2022

Employment status and its associated factors for patients 12 months after intensive care: Secondary analysis of the SMAP-HoPe study

PONE-D-21-23032R1

Dear Dr. Unoki,

We’re pleased to inform you that your manuscript has been judged scientifically suitable for publication and will be formally accepted for publication once it meets all outstanding technical requirements.

Kind regards,

Gebisa Guyasa Kabito, MPH

Academic Editor

PLOS ONE
---

## [Editor Report · Acceptance letter]

28 Feb 2022

PONE-D-21-23032R1 

Employment status and its associated factors for patients 12 months after intensive care: Secondary analysis of the SMAP-HoPe study 

Dear Dr. Unoki:

I'm pleased to inform you that your manuscript has been deemed suitable for publication in PLOS ONE. Congratulations! Your manuscript is now with our production department. 

Kind regards, 

on behalf of

Dr. Gebisa Guyasa Kabito 

Academic Editor

PLOS ONE